# A Cost–Benefit Analysis of COVID-19 Vaccination in Catalonia

**DOI:** 10.3390/vaccines10010059

**Published:** 2021-12-31

**Authors:** Francesc López, Martí Català, Clara Prats, Oriol Estrada, Irene Oliva, Núria Prat, Mar Isnard, Roser Vallès, Marc Vilar, Bonaventura Clotet, Josep Maria Argimon, Anna Aran, Jordi Ara

**Affiliations:** 1Directorate for Innovation and Interdisciplinary Cooperation, North Metropolitan Territorial Authority, Catalan Institute of Health, 08006 Barcelona, Spain; innovacio.mn.ics@gencat.cat (O.E.); irene.olr.12@gmail.com (I.O.); gterritorial.mn.ics@gencat.cat (J.A.); 2Centre for Research in Health and Economics, Pompeu Fabra University, 08002 Barcelona, Spain; 3Fight AIDS and Infectious Diseases Foundation, 08916 Barcelona, Spain; bclotet@irsicaixa.es; 4Comparative Medicine and Bioimage Centre of Catalonia (CMCiB), Fundació Institut d’Investigació en Ciències de la Salut Germans Trias i Pujol, 08916 Barcelona, Spain; mcatala@igtp.cat (M.C.); clara.prats@upc.edu (C.P.); 5BIOCOM-SC, Physics Department, Universitat Politècnica de Catalunya, 08034 Barcelona, Spain; 6North Metropolitan Primary Care Directorate, Catalan Institute of Health, 08006 Barcelona, Spain; nprat@gencat.cat (N.P.); misnard.bnm.ics@gencat.cat (M.I.); rvallesf@gencat.cat (R.V.); mvilar@gencat.cat (M.V.); 7IrsiCaixa AIDS Research Institute, University Hospital Germans Trias i Pujol, 08916 Barcelona, Spain; 8Ministry of Health, 08007 Barcelona, Spain; jmargimon@gencat.cat; 9Catalan Health Service, Ministry of Health, 08007 Barcelona, Spain; dacrsb@catsalut.cat

**Keywords:** cost–benefit analysis, vaccination, COVID-19, health economics, economic appraisal, pharmacoeconomics

## Abstract

(1) Background: In epidemiological terms, it has been possible to calculate the savings in health resources and the reduction in the health effects of COVID vaccines. Conducting an economic evaluation, some studies have estimated its cost-effectiveness; the vaccination shows highly favorable results, cost-saving in some cases. (2) Methods: Cost–benefit analysis of the vaccination campaign in the North Metropolitan Health Region (Catalonia). An epidemiological model based on observational data and before and after comparison is used. The information on the doses used and the assigned resources (conventional hospital beds, ICU, number of tests) was extracted from administrative data from the largest primary care provider in the region (Catalan Institute of Health). A distinction was made between the social perspective and the health system. (3) Results: the costs of vaccination are estimated at 137 million euros (€48.05/dose administered). This figure is significantly lower than the positive impacts of the vaccination campaign, which are estimated at 470 million euros (€164/dose administered). Of these, 18% corresponds to the reduction in ICU discharges, 16% to the reduction in conventional hospital discharges, 5% to the reduction in PCR tests and 1% to the reduction in RAT tests. The monetization of deaths and cases that avoid sequelae account for 53% and 5% of total savings, respectively. The benefit/cost ratio is estimated at 3.4 from a social perspective and 1.4 from a health system perspective. The social benefits of vaccination are estimated at €116.67 per vaccine dose (€19.93 from the perspective of the health system). (4) Conclusions: The mass vaccination campaign against COVID is cost-saving. From a social perspective, most of these savings come from the monetization of the reduction in mortality and cases with sequelae, although the intervention is equally widely cost-effective from the health system perspective thanks to the reduction in the use of resources. It is concluded that, from an economic perspective, the vaccination campaign has high social returns.

## 1. Introduction

The COVID-19 global pandemic has made the development of vaccines necessary to increase the population’s immunity by stimulating the production of antibodies against the infection. As of October 2021, 23 vaccines have been accepted by the competent authorities and 429 are in the testing phase [1]. In most countries, mass vaccination has resulted in a decrease in new cases and, consequently, adverse effects on health (number of deaths, cases with sequelae) and health resources (ICU stays, patients hospitalized, laboratory tests); with a vaccinated population, waves are fewer, less intense and more short-lived [2].

Some economic evaluations have estimated the cost-effectiveness of vaccination, with very favourable results [3,4,5,6,7], suggesting that vaccines against COVID can reduce healthcare costs by up to 60% [8]. Most of these approaches were conducted ex ante and/or using probabilistic models and highlight that the cost-effectiveness of the vaccination strategy depends on the extent of the infection and the vaccinated population exceeding a certain minimum threshold [9,10]. The consensus, then, is that the vaccination strategy against COVID is cost-effective, evidence that is in line with the economic evaluation of other vaccines, which, in Spain, show net savings or favourable cost-effective ratios [11].

In Catalonia (Spain), the pandemic led to an increase in healthcare spending of approximately 20% in 2020, an increase that does not consider the costs of the vaccination campaign, which only began in January of the following year [12]. As in most territories, the vaccination campaign was carried out in phases: certain groups, such as the elderly, essential staff or immunocompromised patients, have been prioritized, according to the risk of catching and transmitting COVID-19 or their economic impact on society. A wide range of resources, such as medical and non-medical staff, communication elements, refrigerators, cars and marquees have been employed, according to the phase. In addition to the intense dedication made during the stages of the highest incidence of the virus, the campaign has led to an extra economic effort in the public health system. In turn, however, this has significantly reduced pressure on healthcare systems [13].

While it is true that there was no alternative to intervention, it is worth comparing its costs with the savings in terms of health impacts and avoided spending, which can quantify the economic returns for both society and the health system because of the aforementioned efforts made during the vaccination process. In this context, this study aims to perform a cost–benefit analysis of the COVID-19 vaccination strategy for Catalonia compared to a baseline in the absence of vaccination, using the social perspectives and that of the National Health System.

## 2. Materials and Methods

### 2.1. Epidemiological Model

The territorial area subject to evaluation is the North Metropolitan Health Region (the most heavily populated district of the greater Barcelona metropolitan area, with a total of 1,986,032 inhabitants, accounting for 25.9% of the total population of Catalonia). The period analyzed is from 1 January 2021, the date on which it can be considered that vaccination began in Catalonia, and 30 September 2021, when the study was conducted. To identify the distribution of cases, hospitalizations, ICU admissions and death by age groups in the absence of the effect of vaccines, epidemiological data observed between 1 September 2020 and 31 December 2020 are used. During this period, it is estimated that the detection of cases was good and constant over time [14]. This assessment assumes that the socioeconomic context and non-pharmacological measures would have been the same in the absence of vaccines. These data are described in Table 1.

According to this approach, the savings generated by vaccination will be equal to the product of the effectiveness of the vaccine against cases, hospitalizations, ICU admissions and deaths, respectively, by the proportion of fully vaccinated individuals, for the incidence reported in the pre-vaccination period in each age group. It is assumed that 1% of cases suffer from some type of sequelae [15].

As can be seen in Figure 1, most of the doses supplied in the North Metropolitan Region were from Pfizer (70%), AstraZeneca (15%), Moderna (12%) and Janssen (3%). Considering the proportions of these vaccines and their reported effectiveness in clinical trials [16,17,18,19] and in the real setting [20], the following ranges of effectiveness were explored: 60–80% for the incidence of cases, 85–90% in the case of hospitalizations and emergencies and 90–95% in the case of deaths. The model was calculated daily in the analysed period. Vaccines are considered to be effective 21 days after administration; the first dose (of double-dose vaccines) is 70% effective compared to the full vaccination and overcoming the disease is considered a first dose of the vaccine (therefore, individuals who have had the virus and been vaccinated with one dose are considered fully vaccinated) [21,22,23].

No third dose was taken during the test period. Vaccination and infection data were obtained from the institutional register in groups from 10 to over 80 years (9 groups) [24]. Figure 1 shows how the vaccination process evolved in the different age groups. The combination of data on the hospitalizations distribution, admissions and deaths by age groups in the comparison period (in the absence of vaccines, September–December 2020) (Table 1), vaccine protection, vaccination by age groups (Figure 1) and the epidemiological data reported during the period 1 January to 20 September 2021 allows for an estimate of the number of cases, hospitalizations, ICUs and deaths avoided, which are shown in Figure 2. To calculate the number of tests (PCR and RAT) saved case data were used, considering that savings in daily tests are proportional to savings in the number of daily cases.

### 2.2. Cost Parameters

To calculate the average unit cost of the vaccination process, a cost analysis of the administrative data of the Territorial Management of the Catalan Institute of Health (the main Primary Care services provider in Catalonia) of the North Metropolitan Health Region was used as an approximation. Over the analysis period, different teams of this provider administered a total of 2,040,642 vaccine doses, 71% of the total doses in this area (2,854,806). According to a review of the literature, and by the consensus of the authors, the following reference prices were used for vaccines: €15, €20, €7 and €3.5/dose for Pfizer, Moderna, Janssen and AstraZeneca, respectively. This unit cost was extrapolated to all the vaccines administered in the territory.

The vaccination campaign had a direct impact on aspects such as human resources and equipment (refrigerators, marquees, furniture, transport of vaccines, vehicle rental, conservation and maintenance, non-medical equipment, needles and syringes, medical equipment, cleaning and security service). While it is true that such costs have been registered, they are not representative of the real cost of vaccination insofar as they are not part of the provider’s regular structural accounting and, therefore, do not include the depreciation of the system as a whole. Therefore, the figure that best approximates the cost of COVID-19 vaccine inoculation was considered to correspond to the cost of any other vaccine, labelled as “Non-urgent nursing care health centre” (code V03PVC0021) in the catalogue of public rates [25]. Similarly, in relation to the expenditure avoided by vaccination, the reference rates of the health service contractor in Catalonia were used [26]. In the case of laboratory tests (PCR and RAT), the costs reimbursed by the healthcare contractor during COVID were used. Health gains were measured in Quality-Adjusted Life Years (QALYs) associated with deaths and cases of long-term morbidity avoided by the vaccination campaign and were monetized according to previous studies [15].

All costs were measured for the year 2021 and reported in euros. A distinction was made between the social perspectives (all observable effects) and the healthcare system (impact on the expenditure of the system). No discount rate was used.

## 3. Results

According to the epidemiological model used, assuming the socioeconomic context and non-pharmacological measures were the same, and depending on the range in the case of effectiveness, vaccination led to a reduction of between 27,000 and 43,000 infections, between 11,000 and 14,500 hospital discharges, between 1700 and 2,200 ICU discharges and between 2600 and 4300 deaths. Between 260,000 and 420,000 PCR tests and between 130,000 and 210,000 RAT tests were also calculated to have been saved. Table 2 shows the economic impacts of these reductions: for the base case, which uses the averages of these ranges (Scenario 1); for the upper threshold (Scenario 2: higher effectiveness of the vaccine); and for the lower threshold (Scenario 3: lower effectiveness of the vaccine).

In relation to the 2,854,806 doses of vaccine that were subject to analysis, and with regard to the base scenario of effectiveness, these results show that 82 doses prevent one infection, 827 doses prevent one death, 224 doses prevent one hospitalization and 1464 doses prevent one admission to the ICU.

The costs are described in Table 3. The total is €137m, of which €37.26m (13.05%) correspond to the 2,854,806 doses that were administered (at a weighted average price of €13.05), and €99.92 m (72%) for the overall cost of human resources and the depreciation of infrastructure and equipment. The total cost per administered dose was calculated to be €48.05.

According to these values, the following results can be inferred: the vaccination campaign generates positive impacts at the social level, amounting in monetary terms to €164.72 (€67.98 from the perspective of the health system) per dose administered (Table 4). Subtracting the cost of vaccination, the benefit was €116.67 and €19.93, respectively. From the perspective of the health system (considering the savings in hospital discharges and ICU units), the benefit/cost ratio is 1.4; if, in addition, the monetization of the reduction in mortality and morbidity (social perspective) is taken into account, this ratio increases to 3.4. These results are robust at the lower and upper threshold of vaccine effectiveness.

## 4. Discussion

As far as the authors are aware, this is the first study to conduct a cost–benefit analysis of the mass vaccination performance based on observational data. The results suggest that vaccination campaigns for COVID-19 may have a high return for both the health care system and society as a whole. In Catalonia, the impact of mass vaccination was highly beneficial in the last waves, avoiding serious cases, deaths and sequelae, and an excessive healthcare and economic strain on the public health system. In view of the difficulties in vaccinating the entire population, these results strengthen the argument in favour of adopting measures which favour the universality of vaccination campaigns, such as the introduction of co-payments for people who decide not to be vaccinated despite the evidence attesting to the safety and benefit of this measure. Extrapolating from the analysed evidence and assimilating the cost structure and the total percentage of people vaccinated by population range, an estimate can be made for the whole of Catalonia and Spain (11,371,928 and 72,594,573 doses administered as of 5 November 2021) [24,25,26,27], accounting for savings of 1327 and 8469 million euros (227 and 1447 million from the perspective of the health system). It seems, therefore, that prioritizing the vaccination campaign has been a very successful strategy in terms of health policy.

### Limitations

This study has several limitations. In relation to the epidemiological impact, it should be borne in mind that conclusive long-term data on the efficacy of COVID-19 vaccines are not yet available. Recent studies suggest that the vaccine provides temporary immunity against infection, while protection against severe cases (hospitalization and death) is maintained [28,29]. For simplicity, this article assumes a case protection value of 70%, considering these various factors and the duration of the study. In any case, the study suggests that a third dose would maintain the balance in the cost–benefit ratio and prolong its positive impacts. Secondly, this model estimates the cases avoided by the direct effect of vaccination. In reality, each potentially avoided case could have resulted in a new transmission chain; therefore, the results presented here show a lower threshold in terms of the number of cases, hospitalizations, ICU admissions, PCR and RAT tests and avoided deaths. In addition, it should be borne in mind that in the comparative period (second half of 2020), compared to the study period (2021), there is a factor of drift in the dominant variants in the territory (alpha and delta) towards higher infectivity variants.

In relation to the economic model, the study also has several limitations. First, macroeconomic impacts such as the savings derived by avoiding the closure of the territory’s economy are not considered. It is likely that in a non-vaccination scenario, limitations in some sectors or limitations regarding mobility would have had to be imposed, which would have entailed an economic loss that should be considered. Second, there is no official source regarding the costs per vaccine dose: the figure used corresponds to a consensus among the authors, based on a literature review. In this sense, the work highlights the lack of transparency of institutions in providing official data. Third, it would be reasonable to adjust the cost for doses that will expire without being administered: in the absence of better approximations, it is observed that 5.6% of purchased doses have not yet been administered [27]. Fourth, the cost to the healthcare system caused by the underdiagnoses arising from mandatory closures has yet to be assessed, which, according to recent studies performed in Catalonia, could be substantial [30,31].

On the other hand, it should be noted that the analysed period has moments of high and low efficiency, depending on the size and type of the vaccination infrastructure and demand. In this interim analysis, it should also be noted that vaccination kinetics were strongly conditioned until early spring according to dose availability. Future research ought to try to identify the vaccination campaign that has had the highest social return.

## 5. Conclusions

The analysis concludes that the mass vaccination campaign against COVID is cost-saving. From a social perspective, most of these savings come from the monetization of the reduction in mortality and cases with sequelae (B/C ratio = 3.4), although the intervention is equally widely cost-effective from the perspective of the health system thanks to the reduction in hospital beds and ICU and number of laboratory tests (B/C ratio = 1.4). These results are robust with respect to different assumptions regarding vaccine effectiveness. It is concluded that, from an economic perspective, the vaccination campaign has high social returns.

## Figures and Tables

**Figure 1 vaccines-10-00059-f001:**
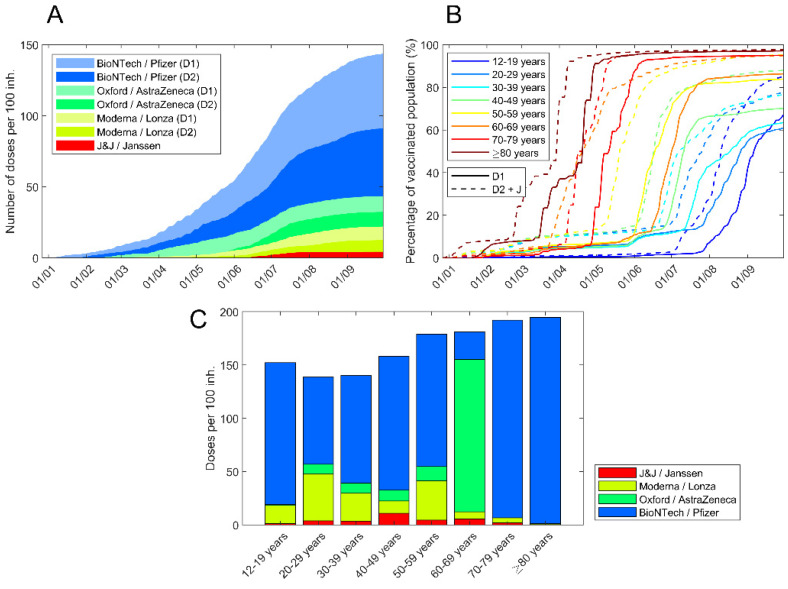
Vaccination process in North Metropolitan Health Region. (**A**) Number of administered doses per 100 inhabitants in the region. Colour according to vaccine manufacturer. Light colours for first dose. (**B**) Vaccination evolution for each age range; first dose is the dashed line. Janssen vaccines are considered a second dose. (**C**) Doses per 100 inhabitants in each age range.

**Figure 2 vaccines-10-00059-f002:**
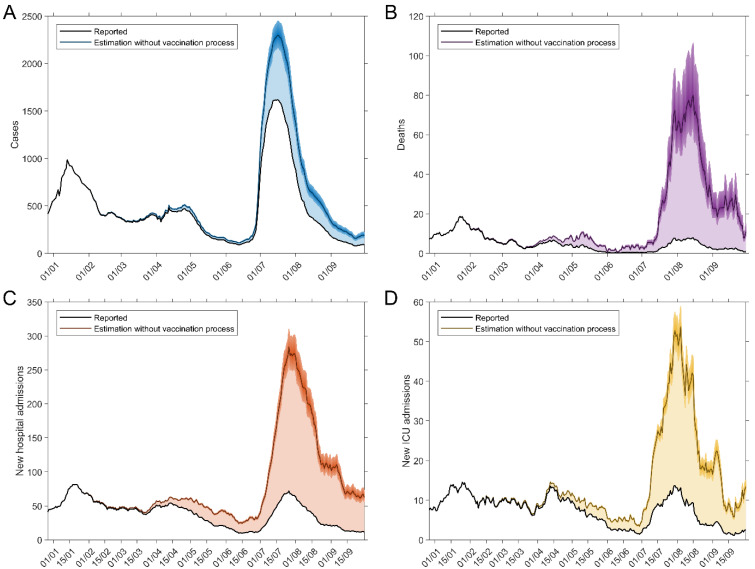
Epidemiological developments with and without the effect of vaccines. (**A**) Cases. (**B**) Deaths. (**C**) New hospitalizations. (**D**) New ICUs.

**Table 1 vaccines-10-00059-t001:** Percentage of cases by age group, in the absence of the effect of the vaccine (epidemiological data 1 September–31 December 2020). North Metropolitan Health Region, Catalonia.

Group	Population	Cases	Hosp.	ICU	Deaths
0–9 years	187,133 (10.0%)	3695 (5.5%)	26 (0.4%)	2 (0.2%)	0 (0.0%)
10–19 years	217,566 (11.6%)	9346 (14.0%)	44 (0.7%)	4 (0.4%)	0 (0.0%)
20–29 years	192,940 (10.3%)	8850 (13.3%)	134 (2.2%)	7 (0.8%)	1 (0.1%)
30–39 years	240,411 (12.8%)	8493 (12.8%)	310 (5.1%)	38 (4.1%)	5 (0.5%)
40–49 years	330,845 (17.6%)	12,005 (18.0%)	604 (9.9%)	89 (9.7%)	11 (1.2%)
50–59 years	268,237 (14.3%)	10,137 (15.2%)	967 (15.8%)	188 (20.4%)	35 (3.7%)
60–69 years	202,241 (10.8%)	6072 (9.1%)	1126 (18.4%)	230 (25.0%)	72 (7.6%)
70–79 years	145,686 (7.8%)	3981 (6.0%)	1246 (20.3%)	260 (28.3%)	173 (18.3%)
over 80 years	93,861 (5.0%)	4009 (6.0%)	1674 (27.3%)	102 (11.1%)	647 (68.5%)

Source: DADESCOVID (Catalonia’s official COVID data. https://dadescovid.cat/. Accessed on 29 December 2021).

**Table 2 vaccines-10-00059-t002:** Benefits of the vaccination campaign. Amounts avoided per scenario.

Perspective		Variable	Unit Cost (€)	N (S1)	N (S2)	N (S3)	€M (S1)	€M (S2)	€M (S3)	€ (%) (S1)
Social	Health System	ICU	43,400/ discharge	1.950	2.200	1.700	85	95	74	18.00%
		Hospita-lizations	6050/ discharge	12.750	14.500	11.000	77	88	67	16.40%
		PCR	75	340.000	420.000	260.000	26	32	20	5.42%
		RAT	40	170.000	210.000	130.000	7	8	5	1.45%
		Deaths	2.92 QALY/death at €25,000/QALY	3.450	4.300	2.600	252	314	190	53.56%
		Cases with sequelae	2.78 QALY/case at €25,000/QALY	350	430	270	24	30	19	5.17%
€ Total saved (millions)							470	567	374	100%

Scenario 1: base model (average); Scenario 2: higher effectiveness of the vaccine; Scenario 3: lower effectiveness of the vaccine.

**Table 3 vaccines-10-00059-t003:** Vaccination campaign costs.

Concept	Cost/Dose (€)	Total Costs (€M)	Cost (%)
HR + Facilities	35.00	99.92	72.84%
Vaccines	13.05	37.26	27.16%
Total	48.05	137	100%

**Table 4 vaccines-10-00059-t004:** Benefit-cost and benefit-dose ratios of the vaccination campaign.

Scenario	B/C Ratio (Social Perspective)	B/C Ratio (Health System Perspective)	Benefit/Dose (Social Perspective)	Benefit/Dose (Health System Perspective)
1 Base; Average effectiveness	3.4	1.4	116.67	19.93
2 Low effectiveness	2.7	1.2	82.81	9.75
3 High effectiveness	4.1	1.6	150.52	30.10

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
