# Peer review of "A Cost–Benefit Analysis of COVID-19 Vaccination in Catalonia"

_vaccines, 2021, doi:10.3390/vaccines10010059_

Round 1
Reviewer 1 Report
This manuscript presents an attempt, one of different strategies for analyzing cost effectiveness, to prove cost-benefit of vaccitanion againts Covid-19 in Catalonia. I appreciate the way the results are shown and I think that what it is presented is interesting. On the other hand the authors describe adequately the intrinsic limits of their study. To render the manuscript more understandable I think it should undergo to an editing of english language and style and a revision of table 2.
Author Response
Dear Reviewer,
Thank you very much for your comments.
Yours sincerely,
The authors
Reviewer 2 Report
This is a very interesting article which I found very useful to increase our knowledge on the Covid-19 pandemic. This economic analysis incorporates real life data from Catalonia, and is not the regular economic modelling assessment, for that reason I see this work of high value.
I'd only provide two comments which I think could improve the current version. The first one is about the persistence of the effectiveness, even it has been already mentioned as one of the limitations of the study, I think a larger effort to estimate what would be the real impact of limiting the effectiveness with time into the analysis will be of benefit of the whole assessment. The latter is because is getting clearer that we'll need a third dose (booster) against Covid-19 so the economic savings are expected to be reduced in the way it has been calculated in this analysis. The second reason to calculate the noted is because not all vaccines have the same rate of persistence, as an example in the US, mRNA vaccines are expected to retain a significant effectiveness around six months and Janssen (one dose) is expected to last 2 months before the US Health Authorities recommend a second dose (booster) for those vaccinated subject, so assuming a constant effectiveness across the study period for all these vaccines could be slightly inflating the potential savings. Thus, any calculation of this mentioned impact will be beneficial for the manuscript (will make the whole exercise more realistic rather that expecting the results significantly). Probably the hardest piece will be to identify the re-vaccination rate (by the way this rate should be mentioned in the current version of the manuscript for those taking the second dose mRNA + Astra Zeneca vaccines - this may differ depending on which vaccines we are talking about).
In addition, I suggest that co-authors consider the appropriate vaccine effectiveness for each vaccine instead of using class average and estimate high and low scenarios. Those scenarios could still be calculated but it sounds reasonable if each vaccine use their own vaccine effectiveness and then the low and high scenarios are done with a closer match to what is the real effectiveness of each vaccine. By doing the latter you can consider the weight of each vaccine in the north Catalonia region and not just an average of all vaccines ("Considering the reported effectiveness of these vaccines in clinical trials [16-19] and in real setting [20], the following ranges of effectiveness have been explored: 60-80% for the incidence of cases, 85-90% in the case of hospitalizations and emergencies and 90-95% in the case of deaths.. The results should be affected dramatically but you may have some more precision.")
Lastly, please double-check the references included within the manuscript, specially the last ones, those may require some rewording (e.g. Reference 27, no accessing date - in comparison to Reference 24 or Reference 26 which doesn't provide much information or Reference 29 which is not including the Journal's volume and pages, etc.)
Author Response
Dear Reviewer,
Thank you very much for your comments, which we address below.
Please note that a calculation has been corrected in table 4, which does not change the meaning of the text nor conclusion.
Yours sincerely,
The authors
----------------------------------
This is a very interesting article which I found very useful to increase our knowledge on the Covid-19 pandemic. This economic analysis incorporates real life data from Catalonia, and is not the regular economic modelling assessment, for that reason I see this work of high value.
I'd only provide two comments which I think could improve the current version. The first one is about the persistence of the effectiveness, even it has been already mentioned as one of the limitations of the study, I think a larger effort to estimate what would be the real impact of limiting the effectiveness with time into the analysis will be of benefit of the whole assessment. The latter is because is getting clearer that we'll need a third dose (booster) against Covid-19 so the economic savings are expected to be reduced in the way it has been calculated in this analysis. The second reason to calculate the noted is because not all vaccines have the same rate of persistence, as an example in the US, mRNA vaccines are expected to retain a significant effectiveness around six months and Janssen (one dose) is expected to last 2 months before the US Health Authorities recommend a second dose (booster) for those vaccinated subject, so assuming a constant effectiveness across the study period for all these vaccines could be slightly inflating the potential savings. Thus, any calculation of this mentioned impact will be beneficial for the manuscript (will make the whole exercise more realistic rather that expecting the results significantly). Probably the hardest piece will be to identify the re-vaccination rate (by the way this rate should be mentioned in the current version of the manuscript for those taking the second dose mRNA + Astra Zeneca vaccines - this may differ depending on which vaccines we are talking about).
----------------------------------
Dear reviewer,
Although this is a very interesting issue, as you mention, it is very difficult to address.
The epidemiological model is developed (and widely published and recognized, as identified in the references) by the Catalan Computational Biology and Bioinformatics Group (BIOCOM-CAT). It is the reference model in the decision-making of the Catalan public health system. In this sense, the approach of this article is very simple, emulating thread by thread (with the BIOCOM authors) the aforementioned model.
As you say, it assumes a weighted average effectiveness depending on the combination of types/brands of vaccine doses administered (observed in the administrative records). However, note that the paper does not assume an "eternal" persistence of effectiveness, only during the period analyzed.
Thank you very much again.
----------------------------------
In addition, I suggest that co-authors consider the appropriate vaccine effectiveness for each vaccine instead of using class average and estimate high and low scenarios. Those scenarios could still be calculated but it sounds reasonable if each vaccine use their own vaccine effectiveness and then the low and high scenarios are done with a closer match to what is the real effectiveness of each vaccine. By doing the latter you can consider the weight of each vaccine in the north Catalonia region and not just an average of all vaccines ("Considering the reported effectiveness of these vaccines in clinical trials [16-19] and in real setting [20], the following ranges of effectiveness have been explored: 60-80% for the incidence of cases, 85-90% in the case of hospitalizations and emergencies and 90-95% in the case of deaths.. The results should be affected dramatically but you may have some more precision.")
----------------------------------
As party replied in our previous comment, we have specified in the text that the effectiveness attributed to vaccines is also due to the weighting of the types of vaccines used (shown by administrative records of the catalan ministry of health).
Thank you very much.
----------------------------------
Lastly, please double-check the references included within the manuscript, specially the last ones, those may require some rewording (e.g. Reference 27, no accessing date - in comparison to Reference 24 or Reference 26 which doesn't provide much information or Reference 29 which is not including the Journal's volume and pages, etc.)
----------------------------------
Done, thank you very much!
----------------------------------